# The Application of Arsenic Trioxide in Ameliorating ABT-737 Target Therapy on Uterine Cervical Cancer Cells through Unique Pathways in Cell Death

**DOI:** 10.3390/cancers12010108

**Published:** 2019-12-31

**Authors:** I-Lun Hsin, Ying-Hsiang Chou, Wei-Li Hung, Jiunn-Liang Ko, Po-Hui Wang

**Affiliations:** 1Institute of Medicine, Chung Shan Medical University, Taichung 40201, Taiwanhideka.chou@gmail.com (Y.-H.C.); ayuatsushi@hotmail.com (W.-L.H.); jlko@csmu.edu.tw (J.-L.K.); 2Department of Medical Imaging and Radiological Sciences, Chung Shan Medical University, Taichung 40201, Taiwan; 3Department of Radiation Oncology, Chung Shan Medical University Hospital, Taichung 40201, Taiwan; 4Department of Obstetrics and Gynecology, Chung Shan Medical University Hospital, Taichung 40201, Taiwan; 5School of Medicine, Chung Shan Medical University, Taichung 40201, Taiwan; 6Department of Medical Research, Chung Shan Medical University Hospital, Taichung 40201, Taiwan

**Keywords:** ABT-737, apoptosis, arsenic trioxide, autophagy, uterine cervical cancer

## Abstract

ABT-737, a B cell lymphoma-2 (Bcl-2) family inhibitor, activates apoptosis in cancer cells. Arsenic trioxide is an apoptosis activator that impairs cancer cell survival. The aim of this study was to evaluate the effect of a combination treatment with ABT-737 and arsenic trioxide on uterine cervical cancer cells. MTT (3-(4,5-dimethylthiazol-2-yl)-25-diphenyltetrazolium bromide) assay revealed that ABT-737 and arsenic trioxide induced a synergistic effect on uterine cervical cancer cells. Arsenic trioxide enhanced ABT-737-induced apoptosis and caspase-7 activation and the ABT-737-mediated reduction of anti-apoptotic protein Mcl-1 in Caski cells. Western blot assay revealed that arsenic trioxide promoted the ABT-737-mediated reduction of CDK6 and thymidylate synthetase in Caski cells. Arsenic trioxide promoted ABT-737-inhibited mitochondrial membrane potential and ABT-737-inhibited ANT expression in Caski cells. However, ABT-737-elicited reactive oxygen species were not enhanced by arsenic trioxide. The combined treatment induced an anti-apoptosis autophagy in SiHa cells. This study is the first to demonstrate that a combination treatment with ABT-737 and arsenic trioxide induces a synergistic effect on uterine cervical cancer cells through apoptosis. Our findings provide new insights into uterine cervical cancer treatment.

## 1. Introduction

B cell lymphoma-2 (Bcl-2) homology 3 (BH3) exhibits pro-apoptotic functions either by directly inducing Bcl-2-associated X protein (Bax) and Bcl-2 homologous antagonist/killer (Bak) or by displacing them away from inhibitory interactions with anti-apoptotic Bcl-2 family members [1]. 4-[4-[(4′-chloro[1,1′-biphenyl]-2-yl)methyl]-1-piperazinyl]-N-[[4-[[(1R)-3-(dimethylamino)-1-[(phenylthio)methyl]propyl]amino]-3-nitrophenyl] sulfonyl]-benzamide (ABT-737) is one of the best-characterized BH3 mimetics. It targets and inhibits anti-apoptotic Bcl-2 family proteins, such as Bcl-2 [2,3]. ABT-737 can induce intrinsic apoptosis pathway and reduce cell viability through Bax and Bak [4]. Cancer cells expressing high Bcl-2 levels are sensitive to ABT-737 treatment [2,5,6], and ABT-737 combines with cytotoxic drugs against solid tumors and hematological malignancy and overcome cancer cell resistance [5,6,7,8,9,10].

Arsenic trioxide (As_2_O_3_) has been applied as a therapeutic agent for more than 2400 years, and its preparation (trade name Trisenox) was approved for the treatment of acute promyelocytic leukemia by the U.S. Food and Drug Administration in September 2000 [11]. It can trigger the intrinsic pathway of apoptosis in human leukemia cells [12]. This apoptotic signaling is presented with oxidative stress, change in mitochondrial membrane potential, and upregulation of apoptotic proteins, leading to cell death. As_2_O_3_ can also induce apoptosis of the cancer cells of uterine cervix via the mitochondrial pathway [13], suppresses the migration and angiogenesis of gastric cancer cells [14], and reverses chemoresistance in hepatocellular carcinoma [15].

Accumulating evidence suggests that the ratio between pro-apoptotic and anti-apoptotic Bcl-2 proteins determines the susceptibility of cancer cells, and cell apoptosis may be induced by inhibiting Bcl-2 [2,3,16,17,18]. A synergistic therapeutic effect on triple-negative breast cancer cell line MDA-MB-231 that overexpresses Bcl-2 can be achieved by combining ABT-737 with docetaxel [10]. In the present study, we investigated a novel treatment strategy that enhances the unique pathways of ABT-737 other than Bax and Bak to determine novel pathways and augment their effects that kill cancer cells in uterine cervixes. In the 2014 Annual Report of Taiwan Cancer Registry, cervical cancer was the second most common type of gynecological cancer and the eighth most frequent cause of female malignancy. It accounted for the age-standardized incidence rate of 8.50 per 100,000 women and a mortality rate of 3.39 in 2014. We hypothesize that the combined treatment of ABT-737 and arsenic trioxide reduces cancer cell viability and enhances the therapeutic effect of both drugs on cervical cancer. The aim of this study was to validate the synergic effect of ABT-737 and arsenic trioxide on cervical cancer cells and develop treatment strategies for this cancer. The pathways through which they promote cancer cell death were determined.

## 2. Materials and Methods

### 2.1. Cell Culture and Chemicals

Caski and SiHa cervical cancer cell lines were obtained from the American Type Tissue Culture Collection. The Caski cells were cultured in Roswell Park Memorial Institute 1640 medium supplemented with 10% heat-inactivated fetal bovine serum (FBS), 0.2% sodium bicarbonate, 2 mM L-glutamine, 100 U/mL penicillin, and 100 μg/mL streptomycin. The SiHa cells were cultured in Dulbecco’s Modified Eagle Medium with 10% FBS, 5.5% sodium bicarbonate, 2 mM L-glutamine, 100 U/mL penicillin, and 100 μg/mL streptomycin. As_2_O_3_, 3-methyladenine (3-MA), and chloroquine were purchased from Sigma (St. Louis, MO, USA). ABT-737 was obtained from Cayman (Ann Arbor, MI, USA).

MTT (3-(4,5-dimethylthiazol-2-yl)-25-diphenyltetrazolium bromide) assay (MTT assay). The Caski and SiHa cells (5 × 10^3^ cells/well) were seeded into a 96-well plate with a 100 μL culture medium. The medium was removed after treatment for 48 h, and 100 μL of the medium containing 0.5 mg/mL MTT (Sigma, M 2128) was added. MTT assay was performed as previously described [19]. The combination index was calculated using the software Compusyn 1.0. [20].

### 2.2. Flow Cytometry for Apoptosis

The Caski and SiHa cells (2 × 10^5^ cells) were seeded onto a 60 mm dish containing 5 mL of culture medium. The medium was removed after 24 h incubation, and 5 mL of fresh medium containing As_2_O_3_ and ABT-737 was added to the dish. The cells were used in investigating apoptosis after 48 h of treatment. Apoptosis was analyzed with fluorescein isothiocyanate (FITC)-labeled annexin V and propidium iodide (PI). A FITC-annexin V apoptosis detection kit I (BD Pharmingen™, San Diego, CA, USA; BD Biosciences, San Jose, CA, USA) was used.

### 2.3. Detection of Mitochondrial Membrane Potential (MMP, ΔΨm)

About 5 × 10^5^ cells CaSki or SiHa cells were seeded onto a 60 mm dish containing 5 mL of culture medium. ABT-737 (2.5 or 5.0 µM) or As_2_O_3_ (2.0 µM) was added 48 hr. At 30 min prior to harvesting, these cells were stained at 37 °C using a 2.5-µM final concentration of 5,5,6,6′-tetrachloro-1,1,3,3′-tetraethylbenzimi-dazolylcarbocy-anine iodide dye (JC-1, T3168, Invitrogen, Carlsbad, CA, USA; Thermo Fisher Scientific, Inc., San Jose, CA, USA) to detect the MMP by fluorescence microscopy and flow cytometry using CellQuest 5.1 software (BD Biosciences, San Jose, CA, USA).

### 2.4. Detection of Reactive Oxygen Species (ROS)

The amount of H_2_O_2_ was determined on the basis of fluorescent light detection for oxidative stress by flow cytometry based on the oxidization and then the conversion of 2′,7′-dichlorodihydrofluorescein diacetate dye (H2DCFDA; Invitrogen; Thermo Fisher Scientific, Inc., San Jose, CA, USA) into fluorescent 2′,7′-dichloro-fluorescein by intracellular ROS. The complete protocols for these analyses are described elsewhere [21,22].

### 2.5. Western Blot Analysis

Anti-cleaved caspase-7 (#9491, Cell Signaling, Danvers, MA, USA), anti-myeloid cell leukemia 1 (#39224, Cell Signaling), anti-Bax (#2772, Cell Signaling), anti-Bak (#12105, Cell Signaling), anti-cyclin-dependent kinase 6 (GTX-103992, Genetex, Irvine, CA, USA), anti-thymidylate synthase (GTX-113289, Genetex, Irvine, CA, USA), anti-voltage-dependent anion channel 1 (Ab14734, Abcam, Cambridge, UK), anti-light chain 3B (#3868, Cell Signaling), anti-adenine nucleotide translocase (SC-9300, Santa Cruz, CA, USA), and anti-β-actin (Sigma, St. Louis, MI, USA) were used to detect the protein expression levels of the indicated genes. The complete protocols for Western blot assay are described in a previous publication [23].

## 3. Results

### 3.1. Synergistic Cell Death Induced by Combing ABT-737 with Arsenic Trioxide as Compared to Single Arsenic Trioxide or ABT-737

MTT assay revealed that the ratios of cell viability of the SiHa cancer cells were 82.8%, 74.0%, and 37.8% after a single treatment of 1.25, 2.5, and 5 μM ABT-737, respectively, and the survival rates after treatment of 0, 2, and 8 μM As_2_O_3_ alone were 100%, 91.8%, and 76%, respectively (Figure 1A). After the combined treatment of 1.25, 2.5, and 5 μM ABT-737 and 2 μM As_2_O_3_, the survival rates of SiHa cancer cells were 69.6%, 40.6%, and 11.4%, respectively; after the combined treatment of 1.25, 2.5, and 5 μM ABT-737 and 8 μM As_2_O_3_, the survival rates were 42.7%, 20.4%, and 12.5%, respectively. The ratios of cell viability of Caski cervical cancer cells after a single treatment of 1.25, 2.5, and 5 μM ABT-737 alone were 96.6%, 69.9%, and 23.7%, respectively, and the survival rates after treatment with 0, 2, and 8 μM As_2_O_3_ were 100%, 69.1%, and 53.7%, respectively (Figure 1B). After the combined treatment of 1.25, 2.5, and 5 μM ABT-737 and 2 μM As_2_O_3_, the survival rates of the Caski cancer cells were 39.3%, 25.6%, and 14.7%, respectively. After the combined treatment of 1.25, 2.5, and 5 μM ABT-737 and 8 μM As_2_O_3_, the survival rates were 26.3%, 18.9%, and 15%, respectively. The combined effects of ABT-737 and As_2_O_3_ were evaluated on the basis of a combination index. In the SiHa and Caski cells, the combined treatment of ABT-737 and As_2_O_3_ manifested synergistic inhibitory effects (combination index of <0.9; Figure 1C,D), and cell viability was observed in nonmalignant cells. This combined treatment did not considerably decrease cell viability in the mouse embryonic fibroblast and human keratinocyte cell line HaCat cells (Appendix A). These results demonstrated that the combined treatment of ABT-737 and As_2_O_3_ induced synergistic inhibitory effects on cervical cancer cells.

### 3.2. Effect of ABT-737 Combined with As_2_O_3_ on Annexin V/PI Assay in Cervical Cancer Cells

Cell death was investigated, and the underlying mechanism was analyzed by annexin V/PI assay. The combined treatment of ABT-737 and As_2_O_3_ increased the population of annexin V(+)/PI(−) and annexin V(+)/PI(+) in the SiHa and Caski cells. This result suggested that ABT-737 and As_2_O_3_ induced apoptotic cell death (Figure 2A). Changes in cleaved caspase-7 after ABT-737 and As_2_O_3_ treatment were observed through Western blot. The combined treatment of ABT-737 and As_2_O_3_ markedly increased cleaved caspase-7 levels in the SiHa cells. Unlike in the SiHa cells, cleaved caspase-7 was slightly upregulated in the Caski cells after the combined treatment as compared with that in separate treatments (Figure 2B). Surprisingly, Z-VAD-FMK, a pan-caspase inhibitor, minimally reversed cytotoxicity in both cells after ABT-737 single agent or combined treatment, but did not reverse cytotoxicity induced by treatment with As_2_O_3_ alone (Appendix A). These results, suggest that SiHa and Caski cells undergo a hybrid form of cell death involving partly apoptosis as well as a non-apoptotic caspase-independent cell death awaiting characterization.

### 3.3. Effect of ABT-737 Combined with As_2_O_3_ on MMP, ΔΨm

JC-1 is a lipophilic mitochondrial agent that detects mitochondrial polarization. JC-1 stains the mitochondria in living cells in a membrane potential-dependent fashion. The so-called J-aggregates, which are favored at a high MMP (mitochondrial membrane potential) and present in the mitochondria, are in equilibrium with JC-1 monomers, which are favored at a low MMP level and present in the cytoplasm [24,25]. The ratio between J-aggregates and monomers was calculated for the analysis of MMP detected by flow cytometry (BD Biosciences, San Jose, CA, USA).

As shown in Figure 3A, MMP level was 7% reduced by ABT-737 in the SiHa cells but not by the combination treatment. Unlike in the SiHa cells, the combined treatment of ABT-737 and As_2_O_3_ markedly reduced MMP level in the Caski cells (Figure 3A). The voltage-dependent anion channel 1 (VDAC1) did not substantially change after the separate treatment of ABT-737 or As_2_O_3_ in the SiHa and Caski cells (Figure 3B,C). ABT-737 decreased As_2_O_3_-induced adenine nucleotide translocase (ANT) upregulation in the SiHa cells (Figure 3B). The amount of ANT was reduced after the separate treatment of ABT-737 in the Caski cells (Figure 3C). Furthermore, ANT reduction was promoted after the combined treatment of ABT-737 and As_2_O_3_ in the Caski cells as compared with that in separate treatments (Figure 3C).

### 3.4. Effect of ABT-737 Combined with As_2_O_3_ on ROS

The amount of H_2_O_2_ was determined on the basis of fluorescent light detection for oxidative stress by flow cytometry using a H2DCFDA dye. The ROS signal shifted to the right in the SiHa and Caski cells after ABT-737 was administered 8 h later. These results indicated that ABT-737 induced oxidative stress in the cervical cancer cells (Figure 4).

### 3.5. Expressions of Anti-Apoptosis Proteins, Cell Cycle Regulated Protein CDK6 and DNA Synthesis TS after ABT-737 and As_2_O_3_ Co-Treatment

ABT-737 alone increased the expression levels of Mcl-1, and As_2_O_3_ alone decreased the ABT-737-induced Mcl-1 (Figure 5A). Unlike in the SiHa cells, ABT-737 alone reduced the expression levels of Mcl-1 in the Caski cells. Furthermore, the combined treatment remarkably decreased Mcl-1 expression in the Caski cells compared with the separate treatments (Figure 5B). The expression levels of Bax and Bak were not altered after the separate or combined treatments in both types of cancer cells (Figure 5A). CDK6 expression decreased after separate ABT-737 or As_2_O_3_ treatments in the SiHa cells (Figure 5C). The amount of TS in the SiHa cells was reduced by the combined treatment of ABT-737 and As_2_O_3_ (Figure 5C). The amounts of CDK6 and TS in the Caski cells progressively decreased as the dose of ABT-737 increased (Figure 5D). After the combined treatment of ABT-737 and As_2_O_3_, the performance of both proteins decreased as the dose increased (Figure 5D).

### 3.6. Co-Treatment of ABT-737 and As_2_O_3_ Induced Anti-Apoptotic Autophagy in SiHa Cells

Autophagy is an anti-apoptosis mechanism [26]. The combined treatment of ABT-737 and As_2_O_3_ increased the LC3B-II/LC3B-I ratio, suggesting that autophagy was induced in the SiHa cells after the treatment (Figure 6A). However, ABT-737 decreased the ratio of LC3B-II/LC3B-I in the Caski cells (Figure 6B). Furthermore, the As_2_O_3_-induced LC3B-II/LC3B-I ratio decreased because of ABT-737 (Figure 6B). To evaluate the role of autophagy as induced by the combined treatment of ABT-737 and As_2_O_3_ in activating apoptosis, we used the autophagy inhibitors 3-MA and chloroquine to inhibit autophagy in the SiHa cells. As shown in Figure 6C, 3-MA partially decreased the LC3B-II/LC3B-I ratio induced by ABT-737 and As_2_O_3_. Compared with the combined treatment of ABT-737 and As_2_O_3_, chloroquine increased the accumulation of LC3B-II, suggesting that the combined treatment induced autophagic flux in the SiHa cells (Figure 6C). Both autophagy inhibitors increased cleaved caspase-7 in the SiHa cells. This result showed that the combined treatment induced anti-apoptotic autophagy in the SiHa cells.

## 4. Discussion

The action of ABT-737 on Bcl-2 has been demonstrated [2,3]. We explored the effects of ABT-737 on cervical cancer from a different perspective. Although the combination of ABT-737 with arsenic trioxide enhances cell apoptosis, their unique actions on cervical cancer cells were explored in other routes rather than the direct inhibition of Bcl-2. The unique actions of ABT-737 and arsenic trioxide on uterine cervical cancer are summarized in Figure 7.

We found that ABT-737 can induce cell death via different pathways other than Bax and Bak in Caski cervical cancer cells. ABT-737 augmented these actions after the combined treatment of As_2_O_3_ with synergistic effect. However, these effects are slightly different in SiHa cells. Aside from the Bax and Bak pathways, other pathways induce cell death in SiHa cells, which are more resistant to chemotherapy than Caski cells because SiHa cells exhibit higher levels of antioxidant enzymes [27]. Therefore, they may have different profiles to cytotoxic drugs. Caski cells, each containing 600 copies of the human papillomavirus 16 (HPV 16) genome, are more sensitive to cytokines than SiHa cells, which only have one or two copies of the HPV 16 genome [28]. Another study reported that SiHa cells present resistance to therapy from their early oncogenic protein HPV E6, causing host cell damage and then promoting cancer cell growth [29]. SiHa cell resistance can be reduced through p53 analogue p73 α overexpression. The present study implied that the stronger resistance of SiHa cells as compared with that of Caski cells may be overcome by the combined treatment of ABT-737 and As_2_O_3_.We have unique findings that ABT-737 can induce cell death via the pathways rather than Bax and Bak in Caski cervical cancer cells and augments these actions after co-treatment of As_2_O_3_ with synergistic effect. However, these effects are slightly diverse in SiHa cells. In addition to via Bax and Bak, other pathways are demonstrated to induce cell death in SiHa cells. SiHa cells were found to be more resistant to chemotherapy than Caski cells because SiHa cells exhibited higher levels of antioxidant enzyme [27]. Therefore, they may have potential to present different profiles to cytotoxic drugs. Caski cells, containing 600 copies of the human papillmovirus 16 (HPV 16) genome per cell were also found to be quite sensitive to the cytokine as compared to SiHa cells, with only one to two copies of the HPV 16 genome [28]. Another study found that SiHa cells present more resistant to therapy results from their early oncogenic protein HPV E6, causing host cells damage and then promoting cancer cell growth [29]. It revealed that SiHa cells resistance can be reduced through the p53 analogue, p73 α overexpression. Our study implied that the stronger resistance of SiHa cells than Caski cells may be overcome via combined ABT-737 and As_2_O_3_, exerting higher Bax and Bak expression and stronger synergistic effect in SiHa cells.

The amount of ANT in the Caski cells was substantially reduced after the combined treatment. This reduction is associated with the dissipation of MMP and the subsequent initiation of cell death. Hexokinase (HK) can bind to the outer mitochondrial membrane (OMM), especially to VDAC1; the outmost portion of the permeability transition pore complex (also known as the mitochondrial permeability transition pore (MPTP)) that is composed of VDAC1 at the OMM; ANT at the inner mitochondrial membrane (IMM); and cyclophilin D in the mitochondrial matrix [30,31]. Hexokinase II can convert glucose-6-phosphate, which is produced by HK in the cytoplasm, to ATP and glucose. ATP is transferred through the VDAC channel at OMM and ANT at IMM to maintain MMP ΔΨ_m_ in aerobic glycolysis, which is the principal process of energy supply for cancer cells [32,33]. When ANT is depleted, MMP is dissipated, and the permeability to MPTP increases [32]. This process leads to cytochrome c release and cell death. A previous study reported that non-small cell lung cancer with ANT overexpression may exhibit resistance to epidermal growth factor receptor/tyrosine kinase therapy [34]. Targeting ANT overcomes the resistance. In the present study, the expression levels of Bax increased after the combined treatment in the SiHa cells but they did not change in the Caski cells. However, Yu et al. found that As_2_O_3_ can trigger apoptosis associated with the dissociation of Bcl-2 from Bax and VDAC and then release cytochrome c from Bax and the VDAC channel that is different from MPTP [13,35]. This process may rescue the lower ANT reduction in SiHa cells as compared with that in Caski cell for the combined treatment of ABT-737 and As_2_O_3_ in our experiments to promote cell death.

We found that Mcl-1 expression in Caski cells obviously decreased after the combined treatment. By contrast, the expression levels of Bax and Bak did not change. Vikström et al. reported that the loss of MCL-1 promotes Bcl-2 inhibition in long-lived plasma cell populations to ABT-737 in animal experiments [36]. Mcl-1 overexpression induces resistance against a number of widely-used anticancer therapies [37], such as paclitaxel [38], vincristine [38], and gemcitabine [39]. However, sensitivity to ABT-737 may be improved through the downregulation of Mcl-1 [40,41]. Furthermore, the combined treatment of ABT-737 and arsenic trioxide remarkably inhibits the xenograft growth of human gastric carcinoma cell lines SGC-7901, synergistically decreases tumor growth, and induces apoptosis in vivo [42]. Therefore, an attractive strategy and promising cancer target for anti-apoptotic Bcl-2 family members using ABT-737 can be applied in combination with additional treatment modalities [43].

Mitotic signals stimulate CDK4 and CDK6 activities, which promote cancer cell proliferation through the entrance of G1 into the S phase in cell cycle [44]. Only ABT-737 cannot arrest cell cycle. However, As_2_O_3_ can induce G2/M arrest by p21 upregulation in acute promyelocytic leukemia [45]. CDK6 decreased after As_2_O_3_ was added for 72 h. Thymidylate synthase participates in cell DNA synthesis [46]. Suppression of TS reduces intracellular thymidine. As_2_O_3_ can reportedly suppress TS in 5-fluorouracil-resistant colorectal cancer cell line HT29 and assumes its use as a chemosensitizer in combination therapy [47]. Therefore, the reduced expression of CDK6 and TS results in cell cycle arrest and defect in DNA synthesis and leads to cell degradation and apoptosis [48].

Autophagy can serve either as a pro-apoptosis or anti-apoptosis mechanism [49]. In the present study, the combined treatment of ABT-737 and As_2_O_3_ increased autophagy in the SiHa cells. Two inhibitors, namely, 3-MA and chloroquine were used to inhibit the different stages of autophagy. The combined treatment induced apoptosis-inhibiting autophagy. Chloroquine treatment increased LC3B-II accumulation in the SiHa cells after the combined treatment, suggesting that ABT-737 and As_2_O_3_ induce autophagic flux. Furthermore, ABT-737 induced apoptosis to a greater extent in the Caski cells than in SiHa cells. These results suggested that autophagy inhibition by ABT-737 increases ABT-737 sensitivity in Caski cells.

## 5. Conclusions

The combined treatment of ABT-737 and arsenic trioxide may exert a synergistic effect to induce the death of cervical cancer cells. The apoptotic ratio in the SiHa and Caski cells substantially increased after the combined treatment. ABT-737 and As_2_O_3_ cause ANT depletion and may simultaneously generate oxidative stress that will lead to MMP reduction [30,33]. These phenomena are sequentially involved in the activation of cleaved caspase-7. Meanwhile, aside from aggravating the above situations, they inhibit the expression of anti-apoptotic proteins and induce cell arrest through CDK6 reduction and interruption of DNA synthesis. Aside from the Bax and Bak pathways, a novel model of how ABT-737 and arsenic trioxide induce cervical cancer cell death in various ways is depicted. These mechanisms can be developed as novel therapeutic strategies for treating uterine cervical cancer.

## Figures and Tables

**Figure 1 cancers-12-00108-f001:**
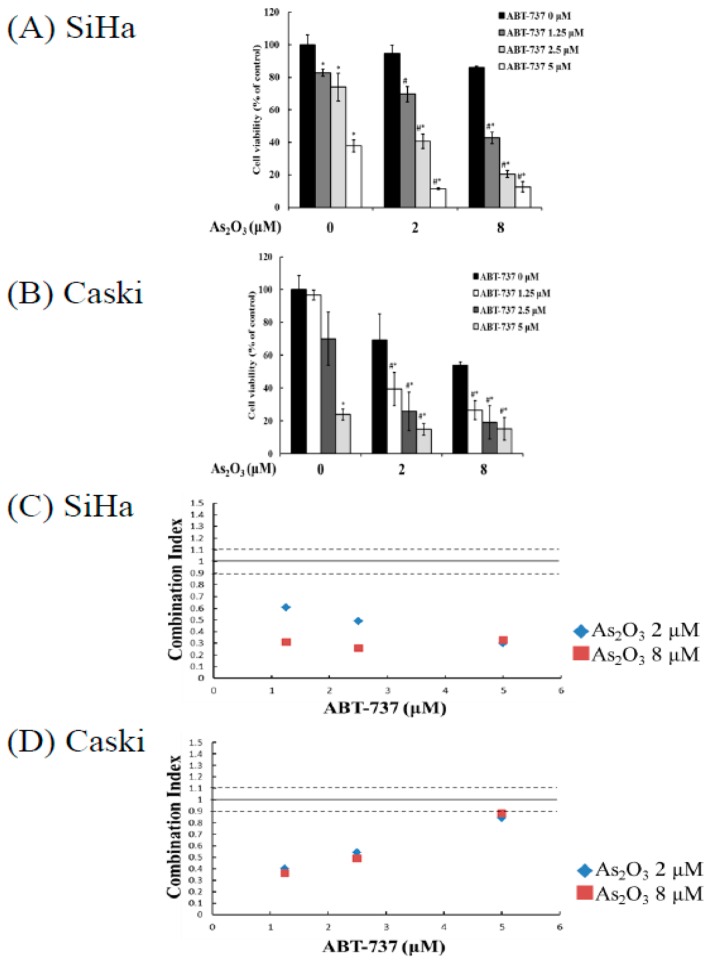
Effects of ABT-737 and As_2_O_3_ on cell viability in cervical cancer cells. (**A**) SiHa (5 × 10^3^ cells/96 wells) were treated with different concentrations of ABT-737 (0, 1.25, 2.5, 5 μM) and As_2_O_3_ (0, 2, 8 μM) for 48 h. (**B**) Caski cells (5 × 10^3^ cells/96 wells) were treated with different concentrations of ABT-737 (0, 1.25, 2.5, 5 μM) and As_2_O_3_ (0, 2, 8 μM) for 48 h. The data were expressed as mean ± SD. Significant differences were determined by one-way ANOVA. * *p* < 0.001, compared with cells without treatment; # *p* < 0.001, compared with A_s_2O_3_ individually treated but no ABT-737 treated cells. (**C**) Combination index of ABT-737 combined with As_2_O_3_ on SiHa cancer cells. (**D**) Combination index of ABT-737 combined with As_2_O_3_ on Caski cancer cells.

**Figure 2 cancers-12-00108-f002:**
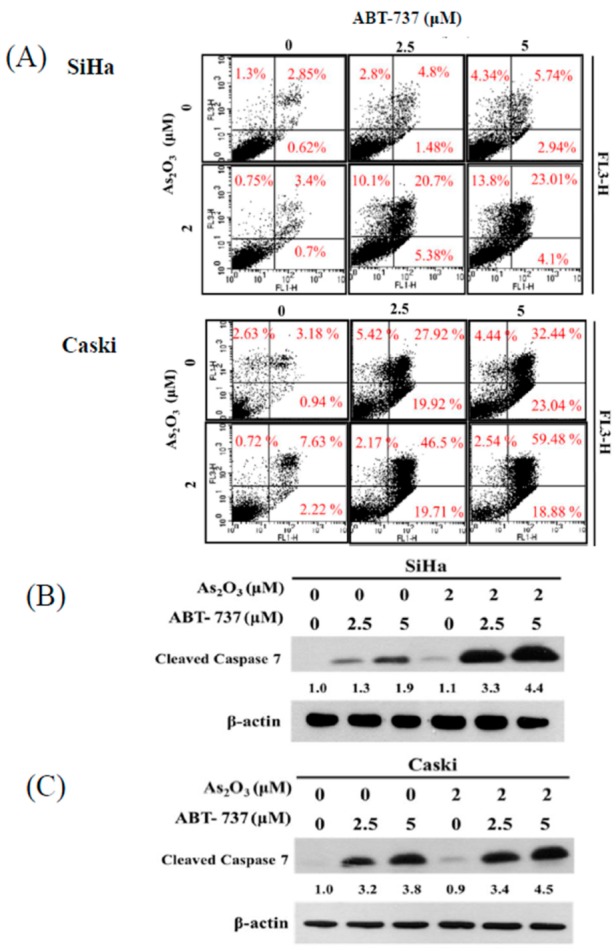
Effects of ABT-737 and As_2_O_3_ mediated apoptosis in cervical cancer cells. (**A**) SiHa and Caski cells (4 × 10^5^ cells/6 cm dish) were co-treated with ABT-737 and As_2_O_3_. The cells were stained with annexin V-fluorescein isothiocyanate (FITC)/propidium iodide (PI) and analyzed by flow cytometry. annexin V-FITC positive (early apoptosis) and annexin V-FITC/PI positive (late apoptosis) were quantified as apoptosis cells. X axis, annexin staining; Y axis, PI staining. (**B**) SiHa and (**C**) Caski cells (4 × 10^5^ cells/6 cm dish) were co-treated with As_2_O_3_ and ABT-737. Cleaved caspase-7 was detected by Western blot. β-actin was as a loading control. The relative ratio of cleaved caspase-7/β-actin is shown.

**Figure 3 cancers-12-00108-f003:**
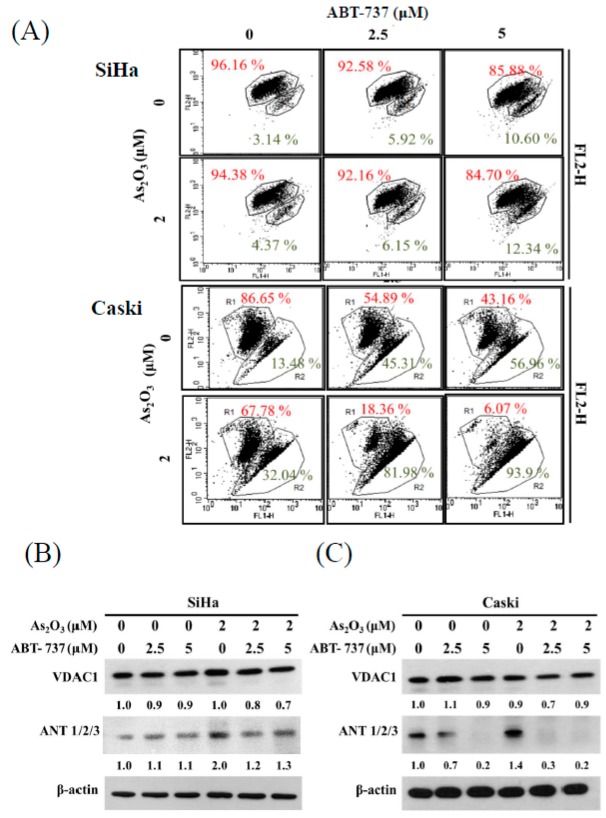
Effects of ABT-737 combined with As_2_O_3_ on mitochondrial membrane potential (ΔΨm) and mitochondrial membrane related proteins. (**A**) SiHa and Caski cells (4 × 10^5^ cells/6 cm dish) were combined with ABT-737 and As_2_O_3_for 48 h. The living cells were stained with 5,5,6,6′-tetrachloro-1,1,3,3′-tetraethylbenzimi-dazolylcarbocy-anine iodide (JC-1) dye to detect the mitochondrial membrane potential by flow cytometry. (**B**) SiHa and (**C**) Caski cells (4 × 10^5^ cells/6 cm dish) were co-treated with ABT-737 and As_2_O_3_ for 48 h. Voltage–dependent anion channel 1 (VDAC1) and adenine nucleotide translocase (ANT) 1/2/3 were detected by Western blot. β-actin was used as a loading control. The relative ratio of VDAC1/β-actin and ANT1/2/3/β-actin are shown.

**Figure 4 cancers-12-00108-f004:**
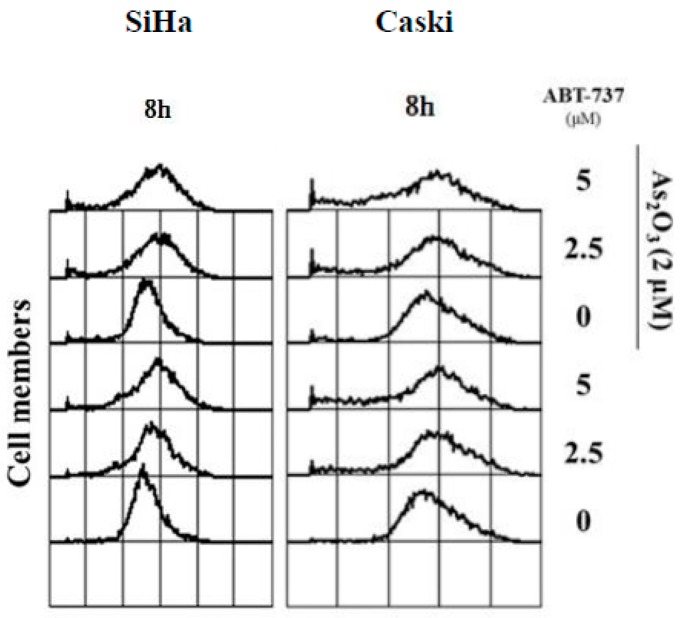
Effects of ABT-737 combined with As_2_O_3_ on reactive oxygen species (ROS) production in cervical cancer cells. SiHa cells (4 × 10^5^ cells/6 cm dish) and Caski (4 × 10^5^ cells/6 cm dish) were co-treated with ABT-737 and As_2_O_3_ for 8 h. After treatment, the cells were stained with 2′,7′-dichlorodihydrofluorescein diacetate (H2DCFDA) dye to detect the production of ROS by flow cytometry.

**Figure 5 cancers-12-00108-f005:**
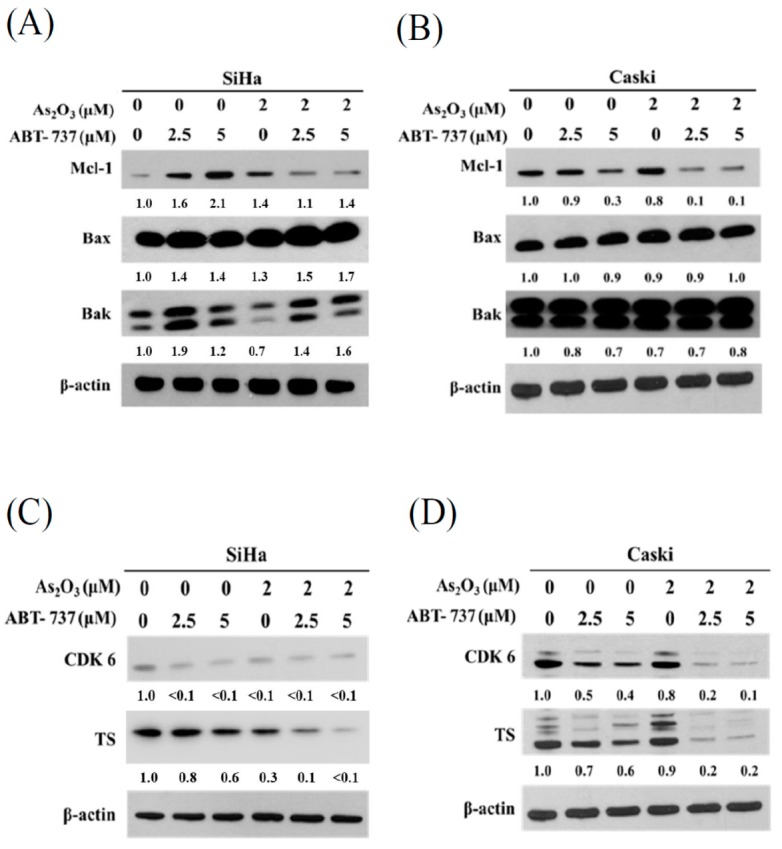
ABT-737 combined with As_2_O_3_ altered expression levels of anti-apoptosis proteins and cell cycle regulating protein. (**A**) SiHa and (**B**) Caski cells (4 × 10^5^ cells/6 cm dish) were combined with ABT-737 and As_2_O_3_ for 48 h. Myeloid cell leukemia 1 (Mcl-1), Bcl-2-associated X protein (Bax) and Bcl-2 homologous antagonist/killer (Bak) were detected by Western blot. β-actin was as a loading control. The relative ratio of Mcl-1/β-actin, BAX/β-actin, and Bak/β-actin were shown. (**C**) SiHa and (**D**) Caski cells (4 × 10^5^ cells/6 cm dish) were combined with ABT-737 and As_2_O_3_ for 48 h. Cyclin-dependent kinase 6 (CDK6) and thymidylate synthase (TS) were detected by Western blot. β-actin was as a loading control. The relative ratio of CDK6/β-actin and TS/β-actin are shown.

**Figure 6 cancers-12-00108-f006:**
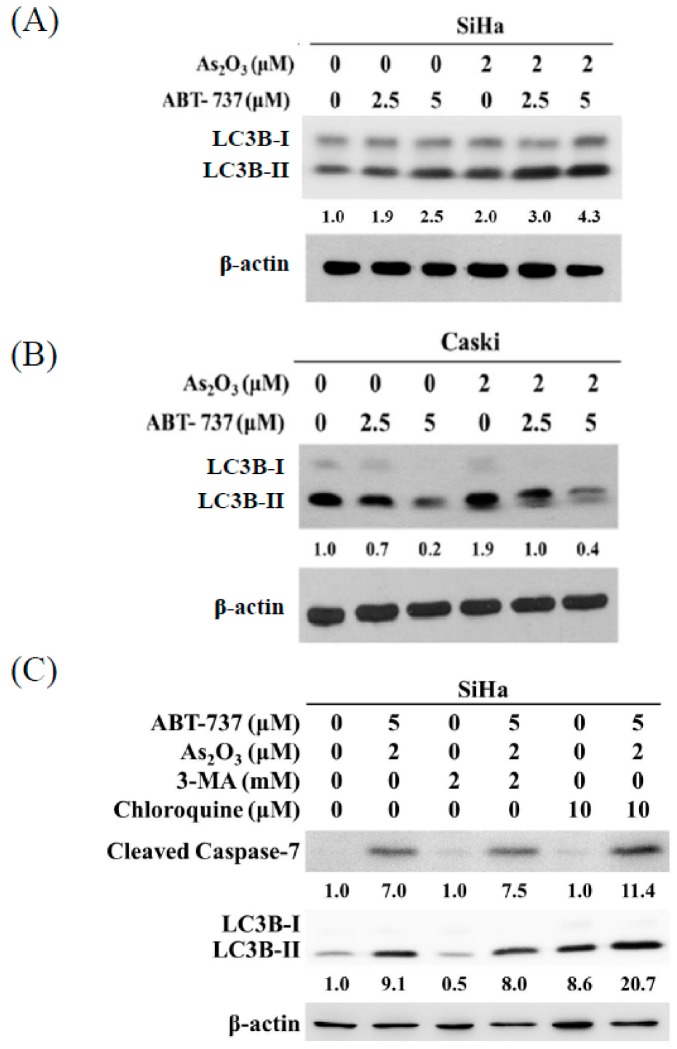
Effects of ABT-737 and As_2_O_3_ on autophagy induction in cervical cancer cells. (**A**) SiHa and (**B**) Caski cells (4 × 10^5^ cells/6 cm dish) were co-treated with ABT-737 and As_2_O_3_ for 48 h. LC3B was detected by Western blot. β-actin was as a loading control. The relative ratio of LC3B II/LC3B I was shown. (**C**) After pretreatment of 3-MA for 1 h, ABT-737, As_2_O_3_, and chloroquine were treated for 48 h. LC3B and cleaved caspase-7 were detected by Western blot. β-actin was as a loading control. The relative ratio of LC3B-II/LC3B-I and cleaved caspase-7/β-actin are shown.

**Figure 7 cancers-12-00108-f007:**
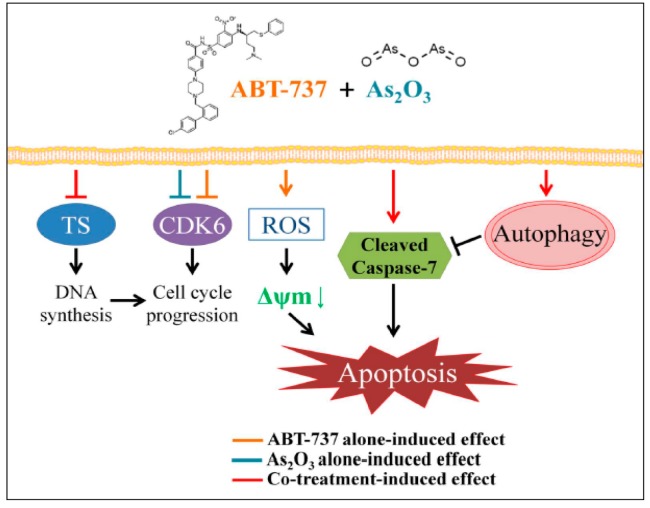
A suggested model of ABT-737 combined with As_2_O_3_ inducing cell death in SiHa cells.

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
