# Peer review of "The Application of Arsenic Trioxide in Ameliorating ABT-737 Target Therapy on Uterine Cervical Cancer Cells through Unique Pathways in Cell Death"

_cancers, 2019, doi:10.3390/cancers12010108_

Round 1
Reviewer 1 Report
The paper is quite more understandable and really improved in the revised version. However, punctation errors are still present: citations are placed following the period of several sentences
Author Response
Thanks for your suggestions. The punctation errors of citation were revised in line 109, 118, 322, 323 and 329.
Reviewer 2 Report
I am unable to access the author’s rebuttal, or perhaps there is no rebuttal addressing the reviewers’ comments. By doing word searches, I noticed that the authors have addressed my point 4, since survivin investigation has been removed in the revised manuscript. Additionally, the authors have done pharmacological tests with caspase inhibitors allowing them to conclude that:
“160 combined treatment as compared with that in separate treatments (Figure 2B). Furthermore, 161 Z-VAD-FMK, a pan-caspase inhibitor, significantly reversed cytotoxicity in the SiHa and 162 Caski cells after the combined treatment (Figure 2S). These results demonstrated that the 163 combined treatment of ABT-737 and As2O3 induced apoptotic cell death in the SiHa and Caski
164 cells.”
However, when I examine Figure S2, there is clearly very little protection with z-VAD-FMK indicating that most of the cell death is non-apoptotic. Therefore, the authors should properly interpret these results which will alter the final conclusions in Figure 7 (in the model forms of cell death should include apoptosis and non-apoptotic caspase-independent cell death). The authors should state that although they observe markers of apoptosis, there is significant caspase-independent non-apoptotic cell death, which may be necrosis, necroptosis, ferroptosis, or other form of cell death that needs to be further established.
I would rewrite lines 160-164 as follows:
“Surprisingly, Z-VAD-FMK, a pan-caspase inhibitor, minimally reversed cytotoxicity in both cells after ABT-737 single agent or combined treatment, but did not reverse cytotoxicity induced by treatment with As2O3 alone (Figure S2). These results, suggest that SiHa and Caski cells undergo a hybrid form of cell death involving partly apoptosis as well as a non-apoptotic caspase-independent cell death awaiting characterization.”
Author Response
Thanks for your suggestions. The sentences were revised (line 160).
This manuscript is a resubmission of an earlier submission. The following is a list of the peer review reports and author responses from that submission.
Round 1
Reviewer 1 Report
In the present paper the authors describe the effect on uterine cervical cancer cells of the combined exposure of ABT-737 and arsenic trioxide. In addition, the investigators also delineated the apoptotic and the autophagic response of the cultured cells to the synergic exposure to the compound.
Major concerns
In the present version of the manuscript, the text displays a considerable number of repetitions, grammatical and punctuation errors throughout the text and in particular in the introduction section. As a consequence, language problems make it hard for a reader to easy understand the information being put across. Thus, an expert in English editing and academic writing is needed to go over the manuscript. In the 3.1 section, authors show, as they state, the synergistic effect of ABT-737/arsenic trioxide induction on SiHa and Caski cells claiming that the combined use of both compounds induced higher level of cell death compared to single compound treatments. Since the MTT is a cell proliferation assay, rather than a cell death assay,to evaluate the cytotoxic effect the investigators should perform more appropriate analysis ( i.e. LDH release assay). In any case, it should have been also shown, as a control, the effect of dose dependent single treatment with arsenic trioxide or ABT-737. In addition, the description of how they calculated either the ratio of cell viability of SiHa (line 129) or the ratio of cell viability of Caski cancer cells (line 134). More importantly, experiments should be performed also on a non-malignant cell line, for example on BJ5ta cells, a human skin fibroblast cell line (see: Albano et al. 2013; Biochimie). The usage of a non-malignant control cell line is certainly necessary because, in a therapeutic perspective, un-specific cell death should represent a serious collateral effect in ubiquitous cell types. In the 3.2 3.3 and 3.4 sections, instead of using terms such as obviously, slightly, gradually…etc. authors should present their results by showing statistical significance, fold increase or decrease, or percentage in order to better validate their results. Results in fig. 5 and 6 should show densitometry of autoradiographs associated to statistical analysis to consistently quantitate level of the markers analyzed. As remarked in point 4, the experiments regarding the analyses of the mitochondrial membrane potential, ROS production and autophagy (3.6 sections), should be performed also in a nonmalignant cell lines.Minor concerns
Statistical analysis should be improved. Authors could use ANOVA and Bonferroni test, instead of the student t-test.
In the text there are references in full text (see lanes: 88, 89 and 90)
Taking everything into account, to make the manuscript acceptable for publication and improve its quality the authors should undertake further work to address all the above points.
Author Response
Major concerns
In the present version of the manuscript, the text displays a considerable number of repetitions, grammatical and punctuation errors throughout the text and in particular in the introduction section. As a consequence, language problems make it hard for a reader to easy understand the information being put across. Thus, an expert in English editing and academic writing is needed to go over the manuscript.Answer: Thanks for your suggestions. The English and writing of manuscript has been revised.
In the 3.1 section, authors show, as they state, the synergistic effect of ABT-737/arsenic trioxide induction on SiHa and Caski cells claiming that the combined use of both compounds induced higher level of cell death compared to single compound treatments. Since the MTT is a cell proliferation assay, rather than a cell death assay, to evaluate the cytotoxic effect the investigators should perform more appropriate analysis (e. LDH release assay). In any case, it should have been also shown, as a control, the effect of dose dependent single treatment with arsenic trioxide or ABT-737.
Answer: Thanks for your suggestions. In Figure 2A, PI(+) population demonstrated that cell death was increased after cotreatment when compared to alone treatment. We suggested this result can provide an evidence of cell death induced by ABT-737 and arsenic trioxide.
In addition, the description of how they calculated either the ratio of cell viability of SiHa (line 129) or the ratio of cell viability of Caski cancer cells (line 134).
Answer: The number of ratio of cell viability was according to the results of MTT.
More importantly, experiments should be performed also on a non-malignant cell line, for example on BJ5ta cells, a human skin fibroblast cell line (see: Albano et al. 2013; Biochimie). The usage of a non-malignant control cell line is certainly necessary because, in a therapeutic perspective, un-specific cell death should represent a serious collateral effect in ubiquitous cell types.
Answer: Thanks for your suggestions. The cell viability of mouse embryonic fibroblast (MEF) and human keratinocyte HaCaT after treating with ABT-737 and arsenic trioxide were analyzed by MTT. The results were shown as follow.
MEF HaCaT
This data was added as Figure S1.
The sentences “This combined treatment did not considerably decrease cell viability in the mouse embryonic fibroblast and human keratinocyte cell line HaCat cells (Figure 1S).” were added Results section (line 138-140, words in red)
In the 3.2 3.3 and 3.4 sections, instead of using terms such as obviously, slightly, gradually…etc. authors should present their results by showing statistical significance, fold increase or decrease, or percentage in order to better validate their results.
Answer: Thanks for your suggestions. The manuscript has been revised.
Results in fig. 5 and 6 should show densitometry of autoradiographs associated to statistical analysis to consistently quantitate level of the markers analyzed.
Answer: Thanks for your suggestions. In Figures 5 and 6, the intensity ratio of each band is shown. We believe that the band intensity ratio provided enough information of level change of the analyzed proteins.
As remarked in point 4, the experiments regarding the analyses of the mitochondrial membrane potential, ROS production and autophagy (3.6 sections), should be performed also in a nonmalignant cell lines.
Answer: Thanks for your suggestions. In point 4, ABT-737 and arsenic trioxide did not induce obviously cytotoxicity in nonmalignant cell lines. We believe that ABT-737 and arsenic trioxide did not markedly alter the mitochondrial membrane potential, ROS production and autophagy in these nonmalignant cells.
Minor concerns
Statistical analysis should be improved. Authors could use ANOVA and Bonferroni test, instead of the student t-Answer: Thanks for your suggestions. Statistical analysis has been performed by one-way anova.
In the text there are references in full text (see lanes: 88, 89 and 90)Answer: Thanks for your correction. The manuscript has been revised.

Reviewer 2 Report
The article entitled “The Application of Arsenic Trioxide in Ameliorating ABT-737 Target Therapy on Uterine Cervical Cancer Cells through Unique Pathways in Cell Death” by Hsin, Wang and colleagues examines combination therapy in model uterine cervical cell lines with regards to apoptosis initiation. While there seems to be synergism between the two drugs in these cell lines, the mechanism of action is confusing and unclear.
Main criticism:
The authors are fishing for potential pathway markers to come up with a story. I feel that the authors need to focus on just one pathway (perhaps intrinsic apoptosis) and provide a more thorough account of the changes in the proteins that regulate it. For instance, a previous publication not discussed by the authors detailed that the BH3-only protein Noxa, Bmf, and Bim were upregulated in As2O3 and that ABT-737 synergized with these proteins in killing multiple myeloma cell lines (see Morales Blood 2008, 111:5152-62)
Immunoblotting has not been performed in the presence of caspase inhibitors to block apoptosis. Therefore, over 48h many cells have lost membrane integrity and also various components that appear expressed at lower levels. The use of caspase inhibitors must be incorporated in every figure.
To test against or for MPTP contributions, the authors could knockout BAK and BAX in one of the cell lines (or upregulate MCL-1, although this is not as clean) to observe the contributions to cell death in ways other than intrinsic apoptosis. This hypothesis will also be supported, one way or another, by the use of caspase inhibitors as suggested above.
Survivin is a known inhibitor of apoptosis that is not implicated in apoptosis, rather being involved in cell division. I think this connection should be excluded in the paper. (See work by Uren et al. Current Biology 2000, 10:1319–1328).
Author Response
Main criticism:
The authors are fishing for potential pathway markers to come up with a story. I feel that the authors need to focus on just one pathway (perhaps intrinsic apoptosis) and provide a more thorough account of the changes in the proteins that regulate it. For instance, a previous publication not discussed by the authors detailed that the BH3-only protein Noxa, Bmf, and Bim were upregulated in As2O3 and that ABT-737 synergized with these proteins in killing multiple myeloma cell lines (see Morales Blood 2008, 111:5152-62)
Immunoblotting has not been performed in the presence of caspase inhibitors to block apoptosis. Therefore, over 48h many cells have lost membrane integrity and also various components that appear expressed at lower levels. The use of caspase inhibitors must be incorporated in every figure.Answer: Thanks for your suggestions. Pan-caspase inhibitor, Z-VAD-FMK, were used to inhibit caspase activation. Using MTT assay, Z-VAD-FMK significantly reversed the As2O3 and ABT-737 induced cytotoxicity. The results were shown as follow.
SiHa Caski
This data was added as Figure S2.
The sentences “Furthermore, Z-VAD-FMK, a pan-caspase inhibitor, significantly reversed cytotoxicity in the SiHa and Caski cells after the combined treatment (Figure 2S).” were added in Results section (line 158-160, words in red)
To test against or for MPTP contributions, the authors could knockout BAK and BAX in one of the cell lines (or upregulate MCL-1, although this is not as clean) to observe the contributions to cell death in ways other than intrinsic apoptosis. This hypothesis will also be supported, one way or another, by the use of caspase inhibitors as suggested above.
Answer: Thanks for your suggestions. In the Figure 5, BAK and BAX did not markedly altered after treatment with As2O3 and ABT-737 in SiHa and Caski cells, suggesting that BAK and BAX did not play an important role in this cotreatment.
Survivin is a known inhibitor of apoptosis that is not implicated in apoptosis, rather being involved in cell division. I think this connection should be excluded in the paper. (See work by Uren et al. Current Biology 2000, 10:1319–1328).
Answer: Thanks for your suggestions. The results of survivin were removed from this manuscript.
